# Tackling Missing Heritability by Use of an Optimum Curve: A Systematic Review and Meta-Analysis

**DOI:** 10.3390/ijms20205104

**Published:** 2019-10-15

**Authors:** Anneke Wegener Sleeswijk, Reinout Heijungs, Sarah Durston

**Affiliations:** 1Department of Psychiatry, University Medical Center Utrecht Brain Center, Heidelberglaan 100, 3584 CX Utrecht, The Netherlands; 2Department of Econometrics and Operations Research, Vrije Universiteit Amsterdam, De Boelelaan 1105, 1081 HV Amsterdam, The Netherlands; 3Institute of Environmental Sciences, Department of Industrial Ecology, Leiden University, Einsteinweg 2, 2333 CC Leiden, The Netherlands

**Keywords:** missing heritability, optimum curve, inverted U, genetic association, context-dependent risk variants, multifactorial variable, 5-HTTLPR polymorphism, autism, systematic review, meta-analysis

## Abstract

Missing heritability is a common problem in psychiatry that impedes precision medicine approaches to autism and other heritable complex disorders. This proof-of-concept study uses a systematic review and meta-analysis of the association between variants of the serotonin transporter promoter (5-HTTLPR) and autism to explore the hypothesis that some missing heritability can be explained using an optimum curve. A systematic literature search was performed to identify transmission disequilibrium tests on the short/long (S/L) 5-HTTLPR polymorphism in relation to autism. We analysed five American, seven European, four Asian and two American/European samples. We found no transmission preference in the joint samples and in Europe, preferential transmission of S in America and preferential transmission of L in Asia. Heritability will be underestimated or missed in genetic association studies if two alternative genetic variants are associated with the same disorder in different subsets of a sample. An optimum curve, relating a multifactorial biological variable that incorporates genes and environment to a score for a human trait, such as social competence, can explain this. We suggest that variants of functionally related genes will sometimes appear in fixed combinations at both sides of an optimum curve and propose that future association studies should account for such combinations.

## 1. Introduction

In this paper, we challenge the well-established finding that there is no association between the short/long 5-HTTLPR polymorphism and autism spectrum disorder (ASD). We show that the short 5-HTTLPR variant is associated with ASD in a well-described, homogeneous subgroup of cases, while the long 5-HTTLPR variant is associated with the same disorder in an alternative homogeneous subgroup and argue that both subgroups may represent the two extreme, biologically opposite ends of an optimum curve. For clinical practice, our findings imply that serotonergic drugs may relieve symptoms of ASD in individuals whose genetic and environmental make-up is represented by one end of an optimum curve, while potentially aggravating the same disorder at the opposite end of the curve. In pharmaceutical research of these medications, beneficial effects may be masked by adverse effects in a mixed ASD population or vice versa, which may cause both under- and overmedication of individuals with ASD. Recent evidence suggests that indeed, a correlation exists between an individual’s 5-HTTLPR variants (short and/or long) and the selective serotonin reuptake inhibitor (SSRI) response in ASD [1]. The concept of an optimum curve can help to distinguish between genetic subgroups. For genetic association research methodology, our findings suggest a new approach to handling statistical heterogeneity. This approach uncovers a type of missing heritability that has not been described before and may be applied in future genetic association studies of complex disorders. Despite the clinical relevance of the complex genetic association between the short/long 5-HTTLPR polymorphism and ASD that we suggest, the primary goal of our study is to use this single case as a proof-of-concept, illustrating a general mechanistic principle that could potentially underlie a yet unknown proportion of missing heritability in any complex disorder.

Unlike somatic diseases, neuropsychiatric disorders are diagnosed on the basis of symptoms alone, and as such, may have multiple aetiologies. A common example is ASD, a neurodevelopmental disorder that is defined by a combination of two core symptom domains: persistent deficits in social communication and social interaction across multiple contexts, and restricted, repetitive patterns of behaviour, interests or activities [2]. Twin and family studies suggest that ASD is largely genetic, with an estimated broad-sense heritability between 64% and 91% [3]. However, none of the genetic ASD susceptibility factors that have been identified to date have proven to be either specific to ASD or universal to all individuals with the condition [4]. Despite substantial effort to identify susceptibility genes through genetic association studies, the genetic substrate of ASD remains unexplained for the majority of cases [5,6]. The gap between estimated and explained heritability of a disorder is sometimes referred to as ‘missing heritability’ [7]. Missing heritability is a common problem in the analysis of complex disorders. Experts on heritability and modes of inheritance have explained the phenomenon by pointing to the complexity of heritability mechanisms and the limitations of current genetic association studies in terms of, among others, the analysis of copy number polymorphic duplications, epistasis, gene-environment interactions, biomolecular interactions that depend on multiple genetic variations, rare combinations and joint small effects of common variants, classes of rare variants that have an effect type in common, translation regulation through non-coding microRNAs, parent-of-origin effects, transgenerational genetic effects and the induction of false negatives through population stratification, when the allelic effect is in the opposite direction to the population effect [8]. Related to this last factor, we hypothesize here that there may be cases of missing heritability that can be explained by a specific combination of context-dependent risk variants [9]. If the two common variants of a gene are both associated with the same disease, but in different genetic or environmental contexts, these genetic associations will mask each other in contextually heterogenic samples. Continental origin is an example of a context that could underlie such effect. Other possible contexts include diet, climate, ethnicity and gender, to mention a few.

One of the earliest and most consistent findings in biological ASD research is the occurrence of elevated levels of whole blood serotonin (5-hydroxytryptamine; 5-HT) in individuals with ASD and in their first-degree relatives [10]. This is likely associated with the activity level of the serotonin transporter (5-HTT or SERT) [11]. The serotonin transporter is encoded by the *SLC6A4* gene on chromosome 17. The promoter region of this gene, the *5-HTT* gene-linked polymorphic region (5-HTTLPR), is characterized by an insertion/deletion polymorphism, resulting in the occurrence of a number of alternative allelic variants. Traditionally, two common variants are distinguished, indicated as the short (484 bp) and the long (528 bp) variant [12]. Functionally, the short variant is associated with a lower transcriptional efficiency—resulting in a lower rate of serotonin uptake by the serotonin transporter—than the long variant. In a number of genetic association studies, a positive association has been found between the occurrence of the short variant and ASD, whereas in other studies, a positive association was identified between the occurrence of the long variant and ASD, and yet other studies report the absence of any statistically significant association. The two available full meta-analyses failed to show any association between either of the 5-HTTLPR variants and ASD, albeit with significant statistical heterogeneity [13,14]. 

The authors of one meta-analysis [13] report overtransmission of the short variant to offspring with ASD in American samples (*p* = 0.002) but not in European and Asian samples. Homogeneity testing in their sample indicated heterogeneity within the set of joint samples, whereas the American and Asian sets were homogeneous. Recent evidence suggests that at the physiological level, ASD can be associated either with enhancement or with depletion of serotonin transporter function and extracellular serotonin availability [15]. Combined, these findings suggest that depletion of extracellular serotonin availability, brought about or enhanced by the short 5-HTTLPR variant, may be a dominant cause of ASD in America but not in Europe or Asia. Considering this apparent geographic stratification, we hypothesize that a difference in continental origin represents a difference in an unknown genetic or environmental factor that collaborates with 5-HTTLPR to affect serotonin metabolism and, with that, social competence. We explored the possibility that both the short and the long 5-HTTLPR variant and an unknown genetic or environmental factor, associated with the continental origin of a subject, differ in their contribution to a multifactorial biological variable (e.g., extracellular serotonin availability) underlying ASD.

In recent literature, dose-response curves between biological variables and psychological traits have often been shown to adopt an inverted U-shape, where the biological ‘dose’ on the horizontal axis will be multifactorial in most cases. The inverted U-shape of these psychological dose–response curves is reminiscent of the well-known physiological optimum curves for enzyme activity as a function of acidity and may reflect a similar mechanism: too high a dose can have the same effect as a deficit. As a matter of course, optima do, in fact, apply to virtually every physiological parameter, ranging from body temperature and blood pressure to the local activities of intra- and extracellular physiological substances. If we plot a measure for general health against each one of these parameters, the resulting graphs will almost always take an inverted U shape (although the U will often be skewed). It is therefore not so far-fetched to presume that the genetic variants that underlie physiological parameters will also relate to optimum curves.

Crucially, we here hypothesize that the mathematical function that relates an explanatory multifactorial variable (e.g., serotonin availability) to the dependent score for social competence is an optimum curve and that ASD is expressed as an insufficient score at the two opposite ends of this curve, where specific combinations of genetic variants and environmental factors that contribute to a multifactorial variable are overrepresented (Figure 1). As a proof of concept, we performed a meta-analysis of 18 transmission disequilibrium test (TDT) studies on the putative association between the short (S) and the long (L) variant of the promoter region 5-HTTLPR of the serotonin transporter gene *SLC6A4* and ASD. Subsequently, we evaluated the results of this meta-analysis in the context of our hypothesis.

Our first research question is whether both the short and the long 5-HTTLPR variant are associated with ASD in two different subgroups of the world-wide ASD population, i.e., in subgroups that differ in continental origin. Our second, more fundamental research question is whether the association between 5-HTTLPR and ASD represents an example of missing heritability that can be explained using an optimum curve. With this proof-of-concept study, we hope to offer a mechanistic approach for addressing missing heritability in ASD and, more broadly, in the pathology of complex disorders.

## 2. Results

### 2.1. Search Results

Our literature search resulted in a set of 17 publications reporting on 18 studies that met our selection criteria [16,17,18,19,20,21,22,23,24,25,26,27,28,29,30,31,32]. Detailed information on the results of the search and selection process, including a completed checklist (Appendix A) and flow diagram (Appendix A) according to the PRISMA (Preferred Reporting Items for Systematic Reviews and Meta-Analyses) protocol [33], is provided in the Appendix A.

### 2.2. Primary Data and Synthesis of the Results

We performed a meta-analysis on the preferential transmission of the short or the long 5-HTTLPR variant from heterozygous parents to offspring with ASD for the full set of 18 TDT studies. Subsequently, we performed separate meta-analyses for three continentally differentiated subsets of studies on samples originating from, respectively, America (five studies), Europe (seven studies) and Asia (four studies). Table 1 provides the data, extracted from the selected studies, along with pooled results and calculated *p*-values on the null-hypothesis of odds ratio (OR) = 1.

In accordance with the results of previous meta-analyses [13,14], we found no association between either the short or the long 5-HTTLPR variant and ASD in the set of joint TDT studies (OR = 1.09; 95% confidence interval (CI) = 0.99 − 1.19; *p* = 0.079). The individual continental meta-analyses yielded a more detailed image: a clear and significant preference for transmission of the short variant to subjects in the American samples (OR = 1.32; CI = 1.14 − 1.52; *p* < 0.001), no preferential transmission in the European samples (OR = 1.02; CI = 0.87 − 1.20; *p* = 0.81), and a clear and significant preference for transmission of the long variant to subjects in the Asian samples (OR = 0.71; CI = 0.56 − 0.91; *p* = 0.007). The odds ratios and their 95% confidence intervals, both sorted and pooled by continent, are presented in a forest plot in Figure 2. The difference in preferential transmission between American and Asian samples is obvious.

### 2.3. Across Study Bias

Table 2 presents the results of homogeneity testing. For the joint set of samples and for European samples, the statistics indicate heterogeneity. However, they clearly point towards homogeneity for American samples (with preferential transmission of the short variant) and for Asian samples (with preferential transmission of the long variant).

A funnel plot [34] suggested that there was no publication bias (see Appendix A). The Egger test [35] confirmed this, as regressing the standardized OR against its precision returned a constant that did not differ significantly from zero (*p* = 0.34) (see Appendix A). The results of a meta-regression of odds ratios on year of publication showed a negligible and non-significant decrease in the odds ratio by approximately −0.002 per year (*p* = 0.95).

### 2.4. Sensitivity Analysis

A sensitivity analysis, based on a stepwise omission of one study at a time, showed the following: within the respective groups of European and American studies, the overall result (no transmission preference in Europe and preferential transmission of the short variant in America) was not affected by leaving out any of the studies, whereas within the group of Asian studies, the exclusion of the study by Cho et al. (2007) [30] changed the association between the long variant and ASD from significant to non-significant (*p* = 0.053).

## 3. Discussion

In this study, we used the concept of an optimum curve to address missing heritability. Furthermore, we used conflicting results and statistical heterogeneity as a proof of concept to reconstruct a concrete biological mechanism.

Our finding that there was no preferential transmission of either the short or the long 5-HTTLPR variant from heterozygous parents to their offspring with ASD at the global scale is in accordance with the results of earlier meta-analyses. In contrast to this overall finding, we found preferential transmission of the short variant to offspring with ASD in America and preferential transmission of the long variant in Asia. These findings support our hypothesis that genetic associations may act through an optimum curve, where the impairment of a trait (i.e., a symptom of a disorder) may result from a multifactorial variable that is either too low or too high. If an optimum curve applies, each variant of a gene may act either as a risk factor or as a protective factor for ASD, depending on the genetic and environmental context in which it occurs. Our physiological interpretation of this phenomenon—that serotonin metabolism can be biased in two opposite ways, each of which may lead to ASD—is in accordance with recent findings [15].

For reasons of simplification, we have restricted ASD to one of its two core symptoms, i.e., ‘deficits in social interaction’, thereby neglecting the other core symptom, i.e., ‘repetitive behaviour or restricted interests’. This begs the question if and how an optimum curve may relate a multifactorial score to a combination of symptoms, rather than to a single symptom only. The answer is that we would need two optimum curves with a common horizontal axis but with different vertical axes: one for the score for social competence, and one for the score for a second psychological trait, i.e., the healthy counterpart of repetitive behaviour or restricted interests.

### 3.1. Relevance

This study is of both clinical and conceptual relevance. The clinical relevance lies in our finding that both the short and the long 5-HTTLPR variants can be associated with autism and that the association is context-dependent. We believe that this finding may help open up the way towards precision medicine for ASD, especially with regard to serotonergic drugs, such as selective serotonin reuptake inhibitors (SSRIs). Although SSRIs are primarily indicated for treating major depression, they have also been prescribed for the treatment of ASD symptoms [36]. SSRIs have been shown to provoke suicidal behaviour in some adolescents, while preventing the same type of behaviour in others [37]. It has been suggested by a number of studies that an association may exist between the response to SSRIs and the short/long 5-HTTLPR polymorphism, with a differential response between Caucasians and Asians [38]. These findings suggest that SSRIs—as an environmental factor that co-determines the variable on the horizontal axis of an optimum curve—might act both as a protective factor and as a risk factor for one and the same disorder, depending on the genetic make-up (or the direction of a serotonergic bias) of the individual using them.

The conceptual relevance of our study lies in the introduction of a new approach to missing heritability, which, as we show, works for at least one case. More broadly, it illustrates the use of mechanistic thinking in addressing missing heritability and the value of unravelling mechanisms to address this problem. 

### 3.2. Strengths and Limitations

The main strength of this study is its mechanistic approach. Although many authors consider population stratification to be a likely source of conflicting results in genetic association studies and a cause of statistical heterogeneity, the phenomenon has not been used to reconstruct a biological mechanism before. However, some limitations must also be acknowledged. First, continental origin is not a biological measure, and its association with a disorder can only be indirect. It may relate to genetic aspects of ethnicity, to cultural environmental aspects such as diet or lifestyle, or to a combination of genetic and environmental aspects. Second, the number of studies included in this meta-analysis is relatively small in light of the ambitious goal of our study. As our sensitivity analysis indicated, this was especially the case for the studies on samples from Asia. Further research is therefore needed to confirm or disprove the genetic associations from our analysis. Third, in the association studies analysed here, the finding that the short/long 5-HTTLPR polymorphism is triallelic rather than biallelic, with two alternative common long variants, L_G_ and L_A_ [39], was not yet taken into consideration. 

### 3.3. Suggestions for Future Research

The results of this study underline that future genetic association studies should focus more often on combinations of genetic variants or on combinations of genetic variants and environmental factors (e.g., continental origin, diet, or psychological stress), rather than on single variants. Groups of functionally related genes (e.g., serotonergic genes) could serve as a starting point for choosing meaningful combinations to assess for possible association with a disorder. 

Continental origin is not a factor that is known to be related directly to serotonin availability. The reason that we chose this factor, rather than a second serotonergic gene, was the fact that the information on it was readily available, combined with the fact that it seemed to correlate with the association between 5-HTTLPR and ASD. Virtually every publication on a genetic association study mentions the continental origin of the assessed samples. Although this information enabled us to perform our proof-of-concept study, we expect combinations of variants of different serotonergic genes to be even more convincingly related to ASD. If it is true that both extreme enhancement and depletion of serotonin availability are associated with ASD [15], it would be expected that combinations of genetic variants that enhance the availability of serotonin reinforce each other’s effect at one end of an optimum curve for social competence as a function of serotonin availability, while variants that deplete its availability do the same at the opposite end. In other words, variants of serotonergic genes may fall apart into two distinct groups. Whereas each group as a whole is associated with ASD, a well-balanced mix of variants from both groups will support optimal social competence. Rather than more literature research, testing this hypothesis requires genetic association studies with an adapted design.

## 4. Methods

### 4.1. Search Strategy

We performed an electronic literature search of the PubMed database at the U.S. National Library of Medicine and of Elsevier’s Embase biomedical and pharmacological database, from inception up to February 2019, using the following combinations of text words as search criteria: autism AND serotonin AND transporter; autism AND SLC6A4; autism AND 5-HTTLPR; autism AND 5HTTLPR. We also manually searched the reference lists of review articles and meta-analyses.

### 4.2. Selection Criteria

Studies were included in our selection if: 1. They were TDT studies on the genetic association between the short/long 5-HTTLPR polymorphism and ASD; 2. The reported data allowed for the calculation of the odds ratio of transmission; 3. Samples did not overlap between studies; 4. The studies were in English. 

### 4.3. Data Extraction

From each study, the following data were extracted independently by the first and the second author: 1. Year of publication; 2. Continental origin(s) of the sample; 3. Sample characteristics; 4. Number of short and long 5-HTTLPR variants, transmitted and not transmitted, respectively, from heterozygous parents to offspring with ASD. Discrepancies were resolved by mutual consent. In cases where published data were unclear, we contacted the authors for clarification.

### 4.4. Statistical Analysis

For every individual study, we calculated the odds ratio. For each odds ratio, the 95% confidence interval was calculated, and the *p*-value corresponding to the null hypothesis OR = 1 was determined through a χ^2^ test. McNemar’s test [40] was used for χ^2^ and *p*-value calculations. 

A pooled odds ratio and its 95% confidence interval and *p*-value were synthesized for all studies combined. We based our meta-analysis on a fixed-effect model rather than on a random-effects model, because our aim was to explicitly investigate the hypothesis of non-random across-study heterogeneity (i.e., through geographical stratification), rather than purely statistical heterogeneity [41].

Heterogeneity among studies was assessed using Cochran’s *Q* [42] and the *I*^2^ statistic [43]. To investigate continental origin as a potential source of heterogeneity, a stratified analysis by continent was conducted through the construction of pooled odds ratios and their 95% confidence intervals and *p*-values for the subgroups of studies that comprised only American, European or Asian samples. To account for potential publication bias, we used Begg’s funnel plot [34], supplemented by the Egger test [35]. Additionally, we performed a meta-regression of odds ratios on year of publication.

A pre-specified sensitivity analysis to evaluate which studies had substantial impact on between-study heterogeneity was performed by leaving out studies one by one, both for the overall group of studies and for each of the continental subgroups of studies separately.

For the statistical analysis, we chose to design our own spreadsheet in Excel 2016 (Microsoft) rather than using an existing software package, in order to provide optimal transparency of the mathematical formulas and calculations used. 

### 4.5. Synthesis of Results

The central measure of this study was the odds ratio, calculated as the odd of transmission of the short relative to that of the long 5-HTTLPR variant, transmitted from heterozygous parents to offspring with ASD. The main outcome measure was preferential transmission, or no preference, for the total set of samples and for subsets of American, European and Asian samples.

## 5. Conclusions

In this proof-of concept study, we used a meta-analysis on 5-HTTLPR and ASD to show how demonstrable genetic associations may be underestimated or missed in conventional genetic association studies. Despite a lack of association between the 5-HTTLPR variant and ASD in a combined set of 18 TDT studies, we found the short variant to be associated with ASD in the subset of American samples and the long variant to be associated with ASD in the subset of samples from Asia. In the joint set, the variants mutually masked each other’s association with ASD. This phenomenon can be explained using an optimum curve, relating a multifactorial biological variable on the horizontal axis, defined by genes and environment, to a score for social competence on the vertical axis. The results support our hypothesis that in a mixed population, two alternative genetic variants of the same gene may both be associated with the same disorder, albeit in different genetic and environmental contexts. This concept may apply to any inverted U-shaped relationship between a multifactorial variable and a physical or psychological trait. If our proof of concept holds up, it has the potential to uncover some of the missing heritability in complex heritable disorders.

## Figures and Tables

**Figure 1 ijms-20-05104-f001:**
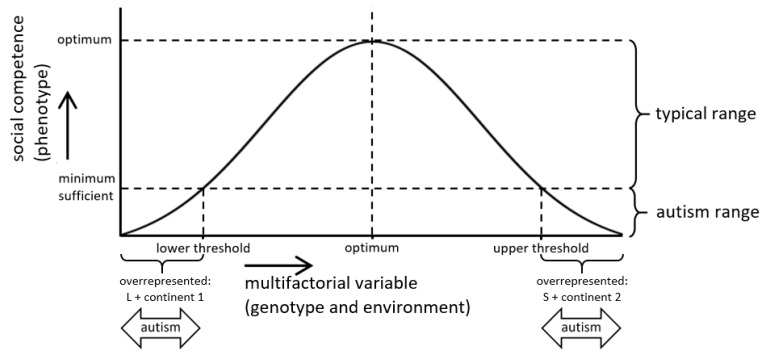
How an optimum curve can relate two alternative genetic variants to the same disorder. A hypothetical optimum curve represents a score for social competence (phenotype) as a function of a multifactorial biological variable (genotype and environment). Both the long (L) serotonin transporter gene-linked polymorphic region (5-HTTLPR) variant and a specific continental origin (continent 1) make a low contribution to this variable, whereas both the short (S) variant and an alternative continental origin (continent 2) make a high contribution. Two separate threshold values (lower and upper) mark the autism ranges of insufficient social competence.

**Figure 2 ijms-20-05104-f002:**
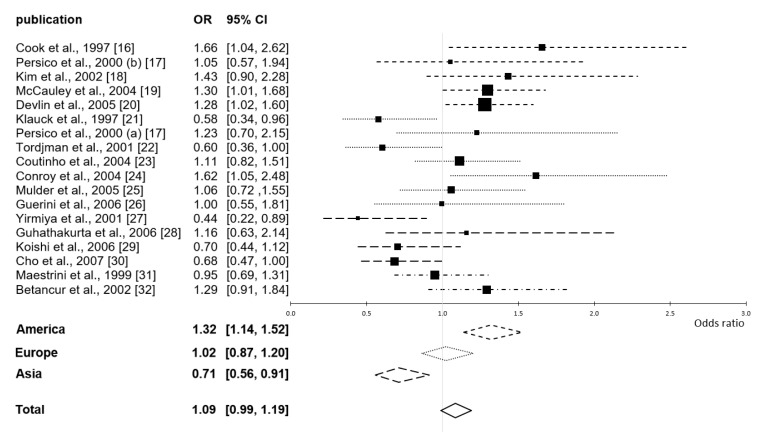
Forest plot of odds ratios (ORs) with 95% confidence intervals (CIs) by continent, representing the odds of transmission of the short relative to those of the long 5-HTTLPR variant, transmitted from heterozygous parents to offspring with ASD. Small dashed lines represent America, dotted lines Europe, large dashed lines Asia, and dot-dashed lines America and Europe together. Both the results of individual studies (above the horizontal axis) and the pooled results (below the horizontal axis) are shown.

**Table 1 ijms-20-05104-t001:** Transmission disequilibrium test (TDT) studies on the genetic association between the short (S) and the long (L) 5-HTTLPR variant and autism spectrum disorder (ASD).

Publication	Origin of Sample	Samples	T: S NT: L	T: L NT: S	Pref.	*p*
	AMERICA					
Cook et al., 1997 [16]	United States	86 trios	48	29	S	0.031
Persico et al., 2000 (sample b) [17]	United States	32 trios 5 multiplex	21	20	–	0.88
Kim et al., 2002 [18]	United States	115 trios	43	30	–	0.13
McCauley et al., 2004 [19]	United States	123 multiplex	135	104	S	0.045
Devlin et al., 2005 [20]	United States	103 trios 125 multiplex	175	137	S	0.031
	EUROPE					
Klauck et al., 1997 [21]	Germany and Austria	65 trios	23	40	L	0.032
Persico et al., 2000 (sample a) [17]	Italy	46 trios 8 duos	27	22	–	0.48
Tordjman et al., 2001 [22]	France	42 trios 2 multiplex	24	40	L	0.046
Coutinho et al., 2004 [23]	Portugal	182 families	88	79	–	0.49
Conroy et al., 2004 [24]	Ireland	84 trios	55	34	S	0.026
Mulder et al., 2005 [25]	The Netherlands	111 trios 3 duos 3 multiplex	54	51	–	0.77
Guerini et al., 2006 [26]	Italy	34 singleton 3 multiplex	22	22	–	1.0
	ASIA					
Yirmiya et al., 2001 [27]	Israel	33 families	11	25	L	0.020
Guhathakurta et al., 2006 [28]	India	61 trios 18 duos	22	19	–	0.64
Koishi et al., 2006 [29]	Japan	104 trios	31	44	–	0.13
Cho et al., 2007 [30]	Korea	126 trios	45	66	L	0.046
	MIXED					
Maestrini et al., 1999 [31]	America and Europe	8 singleton 82 multiplex	72	76	–	0.74
Betancur et al., 2002 [32]	America and Europe	43 trios 53 multiplex	71	55	–	0.15
**total American**		**589 families**			**S**	**<0.001**
**total European**		**588 families**			–	**0.81**
**total Asian**		**342 families**			**L**	**0.007**
**total mixed American/European**		**186 families**				
**total overall**		**1705 families**			–	**0.079**

T, transmitted; NT, not transmitted; pref., preferentially transmitted variant.

**Table 2 ijms-20-05104-t002:** Results of tests for homogeneity of the OR.

Samples	*Q*	*p*	*I* ^2^	Conclusion
American	1.68	0.79	0.00	homogeneous
European	14.27	0.027	0.58	heterogeneous
Asian	4.23	0.24	0.29	homogeneous
All	40.39	0.001	0.58	heterogeneous

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
