# Peer review of "Tackling Missing Heritability by Use of an Optimum Curve: A Systematic Review and Meta-Analysis"

_ijms, 2019, doi:10.3390/ijms20205104_

Round 1

Reviewer 1 Report

In this research paper, the authors developed a mechanistic approach for addressing missing heritability in autism spectrum disorder by analyzing 5-HTTLPR variants and by using the concept of an optimum curve.

- Authors should better address the utility of their method: why this method should provide new information on the clinical relevance? 

- Above all, they should analyze another gene to validate their result.

- It seems that too many variables are associated with the two genetic variants and ASD, while no further improvement is achieved by the method here used.

Author Response

Thank you for your thoughtful remarks, and for expressing your doubts about the relevance of our study. These remarks were extremely helpful, as they made it clear to us that we needed to provide better explanations of why we performed our study, what we did exactly, and above all, of what the role is of our systematic review and meta-analyses in relation to the overarching goal of this study. We feel our paper has significantly benefitted from your comments, and hope you will be satisfied with the changes made and with the text added. Both the direct and the indirect clinical relevance of our findings should now be clearer and much more explicit.

Point 1. Authors should better address the utility of their method: why this method should provide new information on the clinical relevance?

Response 1. We agree that the utility of our method, as well as the reason why it provides new, clinically relevant information, should have been described more clearly. We have adapted the manuscript accordingly.

First, we replaced the concluding sentence of the abstract with a new sentence, that better expresses the core message of our paper.

We suggest that variants of functionally related genes will sometimes appear in fixed combinations at both sides of an optimum curve, and propose future association studies should account for such combinations.

Second, we added a new opening paragraph to the ‘Introduction’ section. In this paragraph, we briefly describe the direct and indirect relevance of our method for clinical practice. The direct clinical relevance relates to medication for ASD – explaining why serotonergic drugs may be helpful for a subgroup of individuals with ASD, while aggravating the condition of others with the same disorder. More insight into this issue by clinicians could help prevent both over- and undermedication. The indirect clinical relevance of our approach is that it can supplement our common endeavours to solve the problem of missing heritability, which would largely add to our understanding of these disorders, and therewith to their treatment. Both the direct and the indirect relevance of our study relate to the development of precision medicine.

In this paper, we challenge the well-established finding that there is no association between the short/long 5-HTTLPR polymorphism and autism spectrum disorder (ASD). We show that the short 5-HTTLPR variant is associated with ASD in a well-described, homogeneous subgroup of cases, while the long 5-HTTLPR variant is associated with the same disorder in an alternative homogeneous subgroup, and argue that both subgroups may represent the two extreme, biologically opposite ends of an optimum curve. For clinical practice, our findings imply that serotonergic drugs may relieve symptoms of ASD in individuals whose genetic and environmental make-up is represented by one end of an optimum curve, while potentially aggravating the same disorder at the opposite end of the curve. In pharmaceutical research of these medications, beneficial effects may be masked by adverse effects in a mixed ASD population, or vice versa, which may cause both under- and overmedication of individuals with ASD. Recent evidence suggests that indeed, a correlation exists between an individual’s 5-HTTLPR variants (short and/or long) and SSRI response in ASD [1]. The concept of an optimum curve can help to distinguish between genetic subgroups. For genetic association research methodology, our findings suggests a new approach to handling statistical heterogeneity. This approach uncovers a type of missing heritability that has not been described before, and may be applied in future genetic association studies of complex disorders. Despite the clinical relevance of the complex genetic association between the short/long 5-HTTLPR polymorphism and ASD that we suggest, the primary goal of our study is to use this single case as a proof-of-concept, illustrating a general mechanistic principle that could potentially underlie a yet unknown proportion of missing heritability in any complex disorder.

Third, we extended the explanation on optimum curves in the ‘Introduction’ section (lines 106 and following), partly to clarify the biological relevance of this concept, and partly to compensate for the removal of text fragment on inverted U shaped curves from the ‘Discussion’ section, that we replaced by a new text fragment on a more urgent discussion point. (See below, ‘Fifth…’.) In particular, we emphasize how common optimum curves are in physiology, which makes it plausible that genetic variants may also be involved in these curves.

In recent literature, dose-response curves between biological variables and psychological traits have often been shown to adopt an inverted U-shape, where the biological ‘dose’ on the horizontal axis will be multifactorial in most cases. The inverted U-shape of these psychological dose-response curves is reminiscent of the well-known physiological optimum curves for enzyme activity as a function of acidity, and may reflect a similar mechanism: too high a dose can have the same effect as a deficit. As a matter of course, optima do, in fact, apply to virtually every physiological parameter, ranging from body temperature and blood pressure to the local activities of intra- and extracellular physiological substances. If we plot a measure for general health against each one of these parameters, the resulting graphs will almost always take an inverted U shape. (Although the U will often be skewed.) It is therefore not so far-fetched to presume that the genetic variants that underlie physiological parameters will also relate to optimum curves.

Fourth, in the ‘Discussion’ section, we moved the paragraph on medication from the subsection Suggestions for Future Research to the subsection Relevance, and made some small adaptations to the text, to make clear how the treatment of individuals with ASD could benefit from our findings. While we originally chose to discuss the implications of our results to medication in a broader sense, we realised that it might be more convincing to apply it more specifically to ASD. In this way, we hope that the clinical relevance of our findings will become more conceivable, especially for the clinicians among our readers.

3.1. Relevance

This study is of both clinical and conceptual relevance. The clinical relevance lies in our finding that both the short and the long 5-HTTLPR variant can be associated with autism, and that the association is context-dependent. We believe that this finding may help open up the way towards precision medicine for ASD, especially with regard to serotonergic drugs, such as selective serotonin reuptake inhibitors (SSRIs). Although SSRIs are primarily indicated for treating major depression, they have also been prescribed for treatment of ASD symptoms [36]. SSRIs have been shown to provoke suicidal behaviour in some adolescents, while preventing the same type of behaviour in others [37]. It has been suggested by a number of studies that an association may exist between the response to SSRIs and the short/long 5-HTTLPR polymorphism, with a differential response between Caucasians and Asians [38]. These findings suggest that SSRIs – as an environmental factor that co-determines the variable on the horizontal axis of an optimum curve – might act both as a protective factor and as a risk factor for one and the same disorder, depending on the genetic make-up (or the direction of a serotonergic bias) of the individual using them.

Fifth, we replaced the remaining part of the subsection Suggestions for Future Research in the discussion section by a new text, in which we focus much more deeply on the primary application possibility of our approach: genetic association studies. Our own research was a meta-analysis, focussing on the admittedly rather peculiar combination of 5-HTTLPR variant and continental origin in relation to ASD. We argue that more sophisticated combinations, such as combinations of the variants of functionally related genes, are expected to be even more clearly associated with disorders. We propose that the design of genetic association studies be adapted, to account for the putative missing heritability that combinations of genetic variants could represent.

3.3. Suggestions for Future Research

The results of this study suggest that future genetic association studies should focus more often on combinations of genetic variants, or on combinations of genetic variants and environmental factors (e.g., continental origin, diet, or psychological stress), rather than on single variants. Groups of functionally related genes (e.g., serotonergic genes), could serve as a starting point for choosing meaningful combinations to assess for possible association with a disorder.

Continental origin is not a factor that is known to be related directly to serotonin availability. The reason that we chose for this factor, rather than for a second serotonergic gene, was the fact that the information on it was readily available, combined with the fact that it seemed to correlate with the association between 5-HTTLPR and ASD. Virtually every publication on a genetic association study mentions the continental origin of the assessed samples. Although this information enabled us to perform our proof-of-concept study, we expect combinations of variants of groups of serotonergic genes to be even more convincingly related to ASD. If it is true that both extreme enhancement and depletion of serotonin availability are associated with ASD [15], it would be expected that combinations of genetic variants that enhance the availability of serotonin reinforce each other’s effect at one end of an optimum curve for social competence as a function of serotonin availability, while variants that deplete its availability do the same at the opposite end. In other words, variants of serotonergic genes may fall apart into two distinct groups. Whereas each group as a whole is associated with ASD, a well-balanced mix of variants from both groups will support optimal social competence. As a consequence, the involved genetic variants will be in linkage disequilibrium within the ASD subpopulation, but not within the general population. Rather than more literature research, testing this hypothesis requires genetic association studies with an adapted design.

Point 2. Above all, they should analyze another gene to validate their result.

Response 2. We do realise that the mere association between the somewhat unexpected combination of 5-HTTLPR variant and continental origin on the one hand and ASD on the other, might be based on coincidence, rather than reflecting a useful, mechanistic association. Moreover, we agree that a single association – even if supported by scientific research – does not suffice to prove the applicability of a newly proposed mechanism. However, the goal of our systematic review and meta-analysis was more modest than that. Rather than proving the applicability, we sought to illustrate the validity of our approach. The concept of an optimum curve as such needs no proof, as it is purely mathematical. As we have now added to the ‘Introduction’ section, optimum curves apply to virtually all physiological mechanisms.

As a matter of course, optima do, in fact, apply to virtually every physiological parameter, ranging from body temperature and blood pressure to the local activities of intra- and extracellular physiological substances. If we plot a measure for general health against each one of these parameters, the resulting graphs will almost always take an inverted U shape. (Although the U will often be skewed.) It is therefore not so far-fetched to presume that the genetic variants that underlie physiological parameters will also relate to optimum curves.

Mathematically, the possibility that optimum curves could also apply to some cases of polygenetic inheritance is self-evident. Whether it also occurs in practice is another question. We foresee that a satisfactory answer to this question cannot be found in the existing scientific literature, simply because it asks for a specific, new approach to genetic association research.

The intriguing question how studies on the association between the 5-HTTLPR polymorphism and ASD could possibly generate such conflicting results inspired us to think of the theoretical possibility of an optimum curve. Subsequently, we used the combination of 5-HTTLPR and continental origin as a proof-of-concept. What we did not intend to do, and cannot do based on this data, is to use this example as a ‘proof-of-applicability’. Our study should therefore be considered a pilot study. Admittedly, the combination of 5-HTTLPR with continental origin is not the most self-evident combination to test for association with ASD. Whether optimum curves do indeed play a significant role in heritability can only be proven by adapted association studies, where meaningful combinations of genetic variants – rather than single variants – serve as genetic unities.

We agree that we had not been clear enough on this issue in the previous draft. Therefore, we have now paid ample attention to this topic in the newly added paragraphs at the end of the ‘Discussion’ section. Not only did we emphasise the function of our systematic review and meta-analysis as a proof-of-concept, but we also explained why validation of the applicability of optimum curves to heritability requires genetic association studies with an adapted design, rather than more literature research on other combinations than 5-HTTLPR and continental origin.

[…] Groups of functionally related genes (e.g., serotonergic genes), could serve as a starting point for choosing meaningful combinations to assess for possible association with a disorder.

Continental origin is not a factor that is known to be related directly to serotonin availability. The reason that we chose for this factor, rather than for a second serotonergic gene, was the fact that the information on it was readily available, combined with the fact that it seemed to correlate with the association between 5-HTTLPR and ASD. Virtually every publication on a genetic association study mentions the continental origin of the assessed samples. Although this information enabled us to perform our proof-of-concept study, we expect combinations of variants of different serotonergic genes to be even more convincingly related to ASD. […] Rather than more literature research, testing this hypothesis requires genetic association studies with an adapted design.

Point 3. It seems that too many variables are associated with the two genetic variants and ASD, while no further improvement is achieved by the method here used.

Response 3. It is unmistakably true that the possible relationship between the 5-HTTLPR polymorphism and ASD has acquired extensive attention in scientific literature for the last two decades, without leading to many convincing results or spectacular new insights in the disorder. Indeed, other, unknown variables may deprive us of a clear view on the true mechanistic relationship, if it exists at all. We can understand that yet another publication on this chewed up subject might be greeted with scepticism, rather than enthusiasm. However, we hope that, with the newly added text in the introduction and discussion sections, we have managed to make clear why we believe that:

we do provide further insight on the relationship between 5-HTTLPR and ASD, in addition to demonstrating a proof-of-concept approach; we needed a chewed up subject to do the latter, as only that provided enough literature on the kind of controversy that we seek to solve; the primary goal of our paper is not to shed more light on the possible association between the 5-HTTLPR polymorphism and ASD (although we are happy to do so), but to shed more light on missing heritability.

Ad 1. The prevailing view in the scientific community is that there is no genetic association between the 5-HTTLPR polymorphism and ASD. Although it is, in a way, satisfying that, with this, a long standing controversary seems to have come to an end, it is also disappointing that the clear physiological findings on specific deviations in serotonin metabolism that are associated with ASD cannot be traced back to their genetic basis. Our findings create a new possibility to find an explanation for these physiological findings, and may, with that, bring us a step closer to treatment options.

Ad 2. Thanks to the fact that the first studies on the genetic association between the 5-HTTLPR polymorphism and ASD produced conflicting results, many more studies followed. Thanks to the availability of so many studies from many parts of the world on the same subject, we could perform a differentiated meta-analysis and use it as a proof-of-concept.

Ad 3. At this stage, it is impossible to estimate the share of missing heritability that is related to optimum curves. However, since it is very common for physiological parameters to show optima, it is quite plausible that the same will be true for the underlying polygenes. Since association studies tend to absorb a large share of the budgets and time spent on the research of complex disorders, it seems wise to address every possible heritability mechanism that might have been overlooked.

Reviewer 2 Report

The authors search for transmission disequilibrium tests and conducted a meta-analysis on the association between the variants of the serotonin transporter promoter and autism in order to explore the hypothesis that missing heritability can be partly explained by the fact that two common variants of a gene can be both associated with a disorder in different genetic or environmental contexts. 

The topic is relevant and the manuscript is well written and clear. Here are some suggestions:

1. The authors might motivate why they decided to use a single database for the systematic review of the literature. 

2. In the Search Results - Details paragraph (supplementary materials) the authors state: "Of the 23-family-based studies (in 22 publications) that we identified through our literature search, five studies did not meet our selection criteria". These numbers seem to be different from the ones reported in the PRISMA flow diagram where the authors report 36 full-text articles assessed for eligibility, 19 excluded, leading to 18 included. 

Please correct numeric errors regarding the number of evaluated/excluded/included articles and also report the number of studies excluded for each specific reason in the "Full-text articles excluded" box of the PRISMA diagram, rather than just the total number of excluded studies. 

3. Did the authors used a fixed-effect or random-effect model to conduct the meta-analyses? 

4. Please report the software used to conduct the meta-analyses

5. At page 6, line 155, the word "shows" should be replaced with "shows"

Author Response

Thank you for going through our paper and the supporting information so thoroughly. Your comments were all very specific, clear, and to the point. We appreciate this, and feel that you added the finishing touch to our systematic review and meta-analysis.

Point 1. The authors might motivate why they decided to use a single database for the systematic review of the literature.

Response 1. This is a valid point that we appreciate you raising. We have now added searches of the Embase database, besides the PubMed database. We found no additional studies to include in our meta-analysis. However, we feel that our study has a sounder basis now. We have adapted the numbers in the PRISMA figure and the text of the supporting information to include the additional studies that the search of the Embase  database provided. We have also mentioned the search of this additional database in the ‘Methods’ section.

Point 2. In the Search Results - Details paragraph (supplementary materials) the authors state: "Of the 23 family-based studies (in 22 publications) that we identified through our literature search, five studies did not meet our selection criteria". These numbers seem to be different from the ones reported in the PRISMA flow diagram where the authors report 36 full-text articles assessed for eligibility, 19 excluded, leading to 18 included. Please correct numeric errors regarding the number of evaluated/excluded/included articles and also report the number of studies excluded for each specific reason in the "Full-text articles excluded" box of the PRISMA diagram, rather than just the total number of excluded studies.

Response 2. There are no numeric errors, but we agree that the text was unclear and confusing at this junction. We have corrected/clarified this.

Out of the 36 full-text articles that were assessed for eligibility, 22 articles reported family-based studies on the genetic association between the short/long 5-HTTLPR polymorphism and autism spectrum disorder (ASD). One of these articles [1] reported two separate family-based studies on samples from two different continents. Hence, we had 23 family based genetic association studies at our disposal. Five of these studies did not meet our selection criteria: the study by Valencia et al. (2012) [2] because it was written in Spanish, the study by Coutinho et al. (2007) [3] because upon request, the authors informed us that the sample of this study was a subset of the sample of a previous study by Coutinho et al. (2004) [4] that was also selected, and the studies by Ramoz et al. (2006) [5], Longo et al. (2009) [6] and Jaiswal et al. (2015) [7] because the reported data did not allow for calculation of the odds ratio (OR) of transmission of heterozygous parents to individuals with ASD of the short relative to the long 5-HTTLPR variant. Therefore, 17 articles, reporting on 18 family based genetic association studies, were included in our meta-analysis. The remaining 19 articles – among which 5 family based and 8 population based association studies  – were excluded.

In the in the ‘Full-text articles excluded’ box of the PRISMA diagram, we now reported the number of studies excluded for each specific reason:

Full-text articles excluded:

not on ASD

(n = 1)

not on 5-HTTLPR S/L

(n = 2)

subgroup analysis

(n = 3)

population based

(n = 8)

not reported in English

(n = 1)

overlapping samples

(n = 1)

OR cannot be derived

(n = 3)

(n = 19)

Point 3. Did the authors used a fixed-effect or random-effect model to conduct the meta-analyses?

Response 3. We used a fixed-effect model. We have now added this in the ‘Methods’ section, and briefly motivated our choice. We have also added a reference on this issue.

We based our meta-analysis on a fixed-effect model, rather than a random-effects model, because our aim was to explicitly investigate the hypothesis of non-random across-study heterogeneity (i.e., through geographical stratification), rather than purely statistical heterogeneity [40].

Point 4. Please report the software used to conduct the meta-analyses

Response 4. We did not use a standard statistical software package to conduct the meta-analysis. Instead, the second author – an associate professor in statistics – designed his own Excel sheet for this purpose. We have now described this in the ‘Methods’ section.

For the statistical analysis, we chose to design our own spreadsheet in Excel 2016 (Microsoft), rather than using an existing software package, in order to provide optimal transparency of the mathematical formulas and calculations used.

Point 5. At page 6, line 155, the word "shows" should be replaced with "show"

Response 5. We have corrected this typo.

Round 2

Reviewer 1 Report

Authors made lots of efforts to improve their manuscript, also according to my comments.

I think in this present form the manuscript is very clear and readable.

Reviewer 2 Report

The authors fully addressed all raised points.